# Precisely timed inhibition facilitates action potential firing for spatial coding in the auditory brainstem

Barbara Beiderbeck[1,2], Michael H. Myoga[1,3], Nicolas I.C. Müller [4], Alexander R. Callan[1,2], Eckhard Friauf [4], Benedikt Grothe[1,3] & Michael Pecka[1]

The integration of excitatory and inhibitory synaptic inputs is fundamental to neuronal processing. In the mammalian auditory brainstem, neurons compare excitatory and inhibitory inputs from the ipsilateral and contralateral ear, respectively, for sound localization. However, the temporal precision and functional roles of inhibition in this integration process are unclear. Here, we demonstrate by in vivo recordings from the lateral superior olive (LSO) that inhibition controls spiking with microsecond precision throughout high frequency click trains. Depending on the relative timing of excitation and inhibition, neuronal spike probability is either suppressed or—unexpectedly—facilitated. In vitro conductance-clamp LSO recordings establish that a reduction in the voltage threshold for spike initiation due to a prior hyperpolarization results in post-inhibitory facilitation of otherwise sub-threshold synaptic events. Thus, microsecond-precise differences in the arrival of inhibition relative to excitation can facilitate spiking in the LSO, thereby promoting spatial sensitivity during the processing of faint sounds.

[1] Department Biology II, Division of Neurobiology, Ludwig-Maximilians-Universitaet Munich, Planegg-Martinsried D-82152, Germany. [2] Graduate School of Systemic Neurosciences, Ludwig-Maximilians-Universitaet Munich, Planegg-Martinsried D-82152, Germany. [3] Max Planck Institute of Neurobiology, Am Klopferspitz 18, Martinsried 82152, Germany. [4] Department of Biology, Animal Physiology Group, University of Kaiserslautern, Kaiserslautern D-67653, Germany. These authors contributed equally: Barbara Beiderbeck, Michael H. Myoga. These authors jointly supervised this work: Benedikt Grothe, Michael Pecka. Correspondence and requests for materials should be addressed to B.G. (email: grothe@lmu.de) or to M.P. (email: pecka@bio.lmu.de)

A crucial function of neuronal circuits is the temporal integration of excitatory and inhibitory inputs[1–5]. While excitatory inputs cause neurons to generate action potentials (APs; "spikes"), inhibitory inputs typically lower neuronal excitability[6–8]. Thus, the particular temporal relationship of the inputs defines a time window for signal propagation and modulation[3,9–12]. This processing motif is particularly prominent in auditory brainstem circuits dedicated to the processing of binaural cues for the localization of sound sources (Fig. 1a)[7,13–15]. In mammals, these cues are the interaural level and time differences (ILD and ITD, respectively) and are first computed by neurons in the lateral and the medial superior olive (LSO and MSO, respectively)[16,17]. In both nuclei, the computation is based on precise interactions of glutamatergic excitation and glycinergic inhibition in response to sounds arriving at the two ears[15,16,18,19] (Fig. 1b, c). A striking shared structural feature is the contralateral inhibitory pathway (Fig. 1b). Here, globular bushy cells in the cochlear nucleus excite glycinergic cells of the medial nucleus of the trapezoid body (MNTB) via highly myelinated and rapidly conducting axons and the giant calyx of Held synapse, specializing this pathway for exquisite speed and reliability[20–22].

Despite these evident specializations for precision, it remains controversial to what extent the precise timing of MNTB-mediated inhibition is a necessary part of the binaural computations[13,14,23–27]. This debate has mainly concerned processing in the MSO, particularly to what extent inhibition shapes spike generation. One MSO in vitro study[24] demonstrated that well-timed inhibitory post-synaptic potentials (IPSPs) can tune ITD-sensitivity by modulating the timing of the net excitatory post-synaptic potentials (EPSPs). The relevance of these mechanisms in the intact brain is, however, discussed controversially[13,23–28], as mechanistic evidence in vivo is lacking. This is partially attributable to the fact that both inhibition and excitation are activated by either ear[29–31], which hinders the interpretation of in vivo data.

In contrast, LSO neurons receive only a subset of the same excitatory and inhibitory inputs, namely excitation from the ipsilateral ear and well-timed MNTB-mediated inhibition from the contralateral ear (Fig. 1b, c)[29–31]. Thus, investigating binaural interactions in the LSO allows a direct assessment of the roles of timed inhibition in synaptic integration and spike generation during spatial processing[19]. Indeed, beyond the well-known gauging of input strength, the comparison of the timing of the excitatory and inhibitory inputs in the LSO is an integral part of ILD computation: any changes in the ILD (i.e., sound source position) entail a change in the relative arrival times of the respective inputs at the LSO[7,32] (Fig. 1d). These timing changes are generated by changes in both the ITD and, more prominently, in latency for the inputs due to the changes in intensity at the ears that are associated with different sound source positions (Fig. 1d). For example, because first-spike latencies can vary by ~1 ms/10 dB any change in the location of a sound source (e.g., from the ipsilateral to the contralateral hemisphere) can cause shifts in the relative timing of inhibition and excitation in the range of many hundreds of µs[7,33].

Thus, ILD processing by LSO neurons consists of the gauging of both input strength (amplitude) as well as timing. The importance of timing is further emphasized by the fact that LSO neurons are also sensitive to ITDs[7,34–38]. However, the LSO has received surprisingly little attention with regard to the cellular mechanisms underlying its sensitivity to input timing. Earlier in vivo studies showed that the time window during which spiking activity is suppressed by inhibition lasts only a few hundreds of microseconds[7,33]. This high temporal precision of inhibition in vivo is coherent with recent in vitro results of EPSCs and IPSCs in the LSO[39] and matches those of IPSPs in the

MSO[15,24,40,41]. Furthermore, synaptic inhibition in the MSO was shown in vitro to even enhance spiking by post-inhibitory facilitation (PIF)[42], and similar properties may be present in LSO neurons[43].

To gain better insight into the functional relevance of precisely timed glycinergic inhibition for binaural spatial processing, we performed single-cell recordings in vivo in the LSO. By disentangling amplitude effects from effects that are specific to input latency, we demonstrate that inhibition sustains microsecond temporal precision throughout high frequency click trains, resulting in input timing-specific modulation of spike timing. Importantly, our data reveal prominent PIF of spiking for a highly specific relative timing between inhibition and excitation. The data suggest that PIF facilitates ILD coding precision of weak excitatory inputs. In vitro whole-cell recordings in mature LSO neurons confirm that these inhibitory functions occur during the binaural integration process and establish that a reduction in the firing threshold due to prior hyperpolarization contributes to PIF. Consequently, we provide direct evidence for the significance of microsecond precise glycinergic inhibition to tune sensitivity to input timing in the auditory brainstem.

## Results

**The role of input timing for binaural processing in the LSO.** To explore the significance of temporal interactions between IPSPs and EPSPs for spatial processing, we recorded spikes from single LSO neurons in anaesthetized gerbils. Changes in the location of free-field sound sources alter the ILD—and hence the input amplitude—at the two ears. This change in the intensity at the ears also results in a change in the relative timing of inputs from the two ears due to two effects (Fig. 1d): first, the different source location changes the external ITD (the relative arrival time of the sound at the two ears). Second, the inputs to the LSO exhibit intensity-dependent changes in conduction latency (Fig. 1d). To be able to study neuronal sensitivity to input timing without confusion of amplitude effects, we presented stimuli with fixed intensities on the ears (the ILD was individually selected for each neuron, see Methods) and experimentally controlled the relative timing of the inhibitory and excitatory inputs. To this end, we introduced "composite timing delays" (cTDs) that replicated the combined timing changes of inputs (ITD and latency changes) that occur normally when altering the sound source position. We used a binaural click-train stimulation paradigm to test the temporal sensitivity of the integration between excitation and inhibition for multiple, repetitive events (Fig. 1e). Clicks invoke robust responses of typically not more than one spike per click and therefore enable assessment of the temporal sensitivity of binaural integration with very high resolution[7,33]. We used trains consisting of six identical clicks, with individual clicks being separated by an inter-click-interval (ICI) varying from 1–5 ms in 1 ms steps (Fig. 1e). This enabled us to not only determine the input timing sensitivity for single integration events at the onset of the stimulus, but also to monitor how this sensitivity changes for later clicks in the train. By varying the ICI in the click-trains, we further assessed the functional time course of the inputs. Figure 1f (left-hand panel) depicts mean response rates from a representative recording of a LSO neuron to 20 repetitions of the click train presented to the excitatory ear only. The probability of the neuron to respond with spikes to monaural clicks was high early in the train but decreased for later clicks (Supplementary Fig. 1a). During binaurally presented click-trains, the response rates of the same neuron were clearly modulated by the cTD, i.e., the relative timing of the inputs from the two ears (Fig. 1f, right-hand panel). As expected from an addition of inhibitory input, minimum response rates ("min-rates") fell below the average

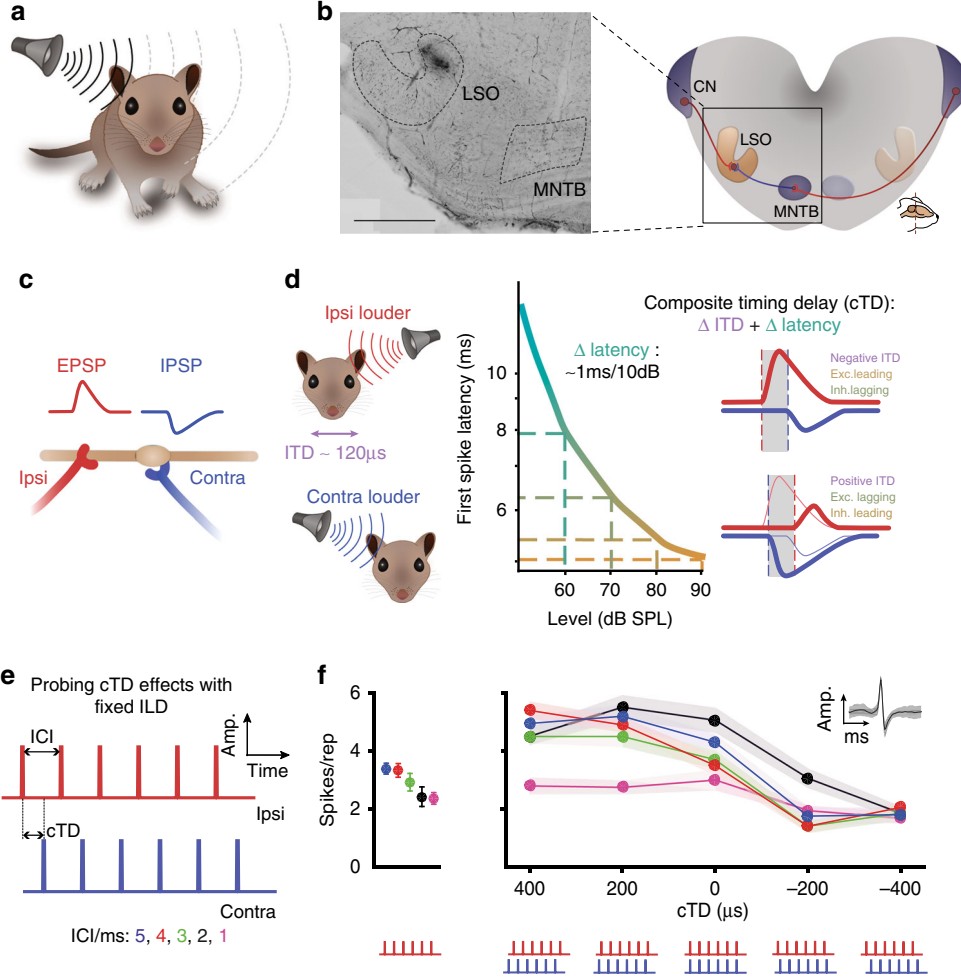

**Fig. 1** ILD processing in the LSO requires high neuronal sensitivity to binaural input timing. **a** The acoustic head shadow produces location-specific interaural level differences (ILDs) between the ears. **b** Right: ILDs are first detected by LSO neurons by comparing ipsilateral excitatory input from the cochlear nucleus (CN) and contralateral inhibitory input from the medial nucleus of the trapezoid body (MNTB). Left: the recording site at the medial limb of the LSO for data shown in **f** was labeling with horseradish-peroxidase, which was deposited via the recording electrode. Scale bar: 500 μm. **c** During ILD computation, LSO neurons integrate the relative strength (amplitude) and timing of inhibitory and excitatory post-synaptic potentials (IPSPs and EPSPs, respectively). **d** Changes in the sound source location cause significant changes in the relative timing of the inputs to the LSO: higher absolute sound intensity cause PSPs with larger amplitude, and also shorter latencies. The graph (middle panel) illustrates these level-dependent changes in first spike latency of an auditory nerve fiber (adapted from[63]). Together with the location-specific ITD (in the range of <120 μs), these latency changes largely determine the range of input timing changes associated with changes in the location of a sound source (right panel). Thus, ILD computation also involves gauging of the relative timing of EPSPs and IPSPs. **e** The acoustic stimulus consisted of six clicks (50 μs duration per click), spaced at five different ICIs (1–5 ms in 1 ms increments). For binaural presentation, the cTD was varied between −400 and +400 μs in 200 μs steps (negative values denote ipsilateral ear leading stimuli). **f** Left: Mean spike rates per repetition from an example neuron (CF: 24.3 kHz) during (ipsilateral) excitatory only click train stimulation. (74 dB SPL, 20 repetitions each, ICIs color-coded as denoted in **e**. Right: cTD-response functions of the same neuron (ipsi: 74 dB SPL, contra: 79 dB SPL, 20 repetitions). Inset shows mean spike waveform. Filled circles and shaded areas denote mean and standard error of the mean (s.e.m.). Note that maximal response rates at +200 and +400 μs cTD exceed mean rates during excitatory only stimulation

monaural rate for all ICIs tested. Spike rate reduction occurred at cTDs that were slightly leading at the ipsilateral ear (the "min-cTD"), and conversely, peaked for cTDs of 200 to 400 μs leading on the contralateral ear (the "max-cTD"). Unexpectedly, however, the maximum binaural response rates ("max-rates") in this neuron exceeded the respective monaural response rate at specific cTDs for all ICIs (compare left and right panels in Fig. 1f).

We observed similar cTD-dependent modulation of responsiveness between monaural and binaural stimulation across the sample (Fig. 2, $n = 17$ neurons from 12 animals, characteristic frequencies [CFs] 15–36 kHz, see Supplementary Fig. 1). Average response rates during stimulation of the excitatory ear only were significantly altered by binaural stimulation. Specifically, both the ICI and the cTD of the clicks had a significant influence on spike

rates [two-way analysis of variance (ANOVA), $P(\text{ICI}) < 0.0001$, $F(4, 64) = 22.48$; $P(\text{cTD}) < 0.0001$, $F(2, 32) = 16.16$; Interactions: $P < 0.0001$, $F(8, 128) = 9.59$]. As expected for the stimulation of the inhibitory ear, min-rates for all ICIs were consistently lower compared to excitatory only stimulation (Fig. 2a). Conversely, average max-rates were higher at all ICIs except for 1 ms (Fig. 2 a). This unexpected increase in spike rate raises the possibility of activating a (unknown) contralateral excitatory input to the LSO that influences spike rates. To control for this possibility, we tested the effect of increasing the sound level on the contralateral ear by +5 dB (ipsilateral level un-changed). Increasing level resulted in a significant decrease in maximal spike rates (Fig. 2b, see legend for details), which is in accordance with a purely inhibitory projection from the contralateral ear to the LSO. This

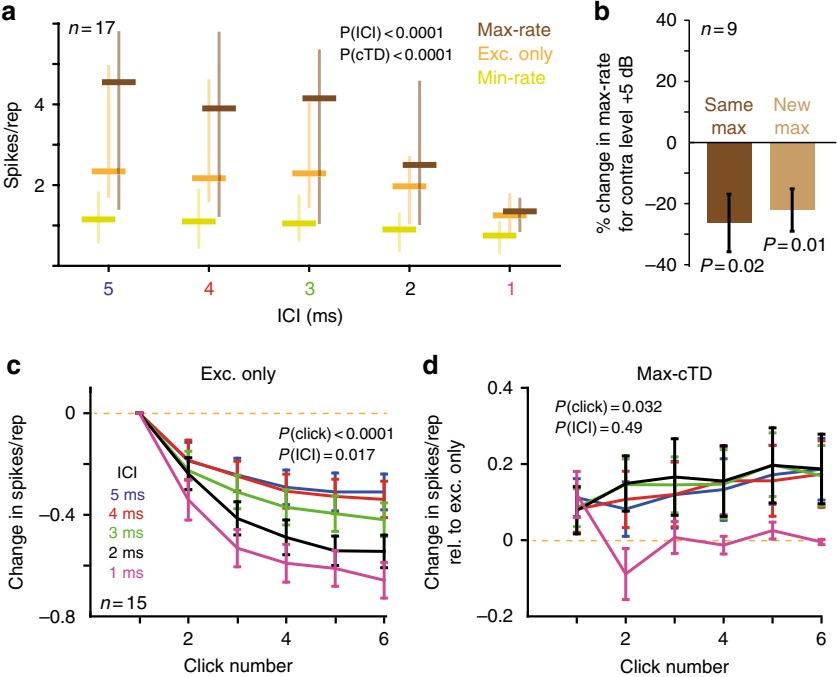

**Fig. 2** Inhibition enhances spiking throughout the click train. **a** Sample response rates ($n = 17$ neurons, median and interquartile range) to click-trains at different ICIs are shown during excitation only (orange), at min-cTD (yellow) and at max-cTD (brown). Both the ICI and the cTD of the click train significantly modulated spike rates (P-values derived from two-way ANOVA test). **b** Increasing the level on the contralateral ear (ipsilateral level unchanged) reduced the max rates, both if measured at the max cTD for the original ILD (dark brown, mean change ± s.e.m.: $-26.34 \pm 9.4\%$, $P = 0.02$, t-test, $n = 9$ neurons) and at the max cTD for the new ILD (light brown, $-22.11 \pm 6.9\%$, $P = 0.01$, t-test, $n = 9$ neurons). **c** Single-click spike rate analysis ($n = 15$): Normalized mean spike rates (normalization to mean response rates to the first click) decreased as a function of the individual clicks and ICI (two-way ANOVA, $P$(click) < 0.0001, $F_{(5, 400)} = 124.1$; $P$(ICI) = 0.017, $F_{(4, 80)} = 3.2$). Interactions between clicks and ICI were also significant ($P < 0.0001$, $F_{(20, 400)} = 2.76$). **d** For ICIs >1 ms, the mean response rates at max-cTD were significantly higher than Exc. only response rates across all clicks (two-way ANOVA, $P$(click) = 0.032, $F_{(5, 400)} = 2.47$; $P$(ICI) = 0.49, $F_{(4, 80)} = 0.87$)

finding has two critical implications regarding the increased spiking at max cTD: first, it was not caused by a contralaterally driven excitation. Second, spike enhancement is specific to a limited range of ILDs.

The unexpected effect of contralateral inputs enhancing spiking in a timing-specific manner may not be equally pronounced for all spikes elicited during a click train. Specifically, spikes elicited by later clicks may be affected differently than early spikes because of the relative time course of inhibition and potential summation effects. To test this, we analyzed response dynamics for each click individually by binning spikes according to their occurrence (Fig. 2c, d; $n = 15$ neurons from 11 animals, see Methods). During monaural stimulation (excitation only), a progressive decline in spike rate probability to clicks later in the train with decreasing ICI became apparent (Fig. 2c, two-way ANOVA, $P$(click) < 0.0001, $F_{(5, 400)} = 124.1$; $P$(ICI) = 0.017, $F_{(4, 80)} = 3.2$; interactions: $P < 0.0001$, $F_{(20, 400)} = 2.76$). Interestingly, this decline was partially counteracted by the presence of inhibitory inputs during binaural stimulation at max-cTDs (Fig. 2d): the mean max-rate per click increased relative to rates elicited during monaural stimulation, particularly for clicks later in the train (two-way ANOVA, $P$(click) = 0.032, $F_{(5, 400)} = 2.47$; $P$(ICI) = 0.49, $F_{(4, 80)} = 0.87$; Interactions: $P = 0.28$, $F_{(20, 400)} = 1.16$). Thus, dependent on the relative timing, contralateral inputs in the LSO decreased or facilitated spiking to excitatory inputs in the LSO.

**Preceding inhibition facilitates spiking in vivo.** What mechanism could account for increased spiking by addition of

inhibition? Our results so far are reminiscent of facilitatory effects of timed inhibition that have been observed in slice recordings from juvenile MSO[42]. Those in vitro data demonstrated the occurrence of spike facilitation in a temporally precise manner when inhibition was leading excitation. We therefore sought to determine whether a similar temporal relationship also underlies the facilitated responsiveness in LSO in vivo. To this end, we first re-centered the cTD-spike rate function of each neuron to the min-cTD (min-cTDs were widely distributed across the cTD range, Supplementary Fig. 1d). This procedure allowed us to quantify the relative timing of the ipsilateral and contralateral inputs, because the min-cTD can be regarded as the relative time point of functional coincidence between excitation and inhibition[19,34,44] (Fig. 3a), i.e., the time point at which inhibition was maximally effective in suppressing spiking ($\Delta t = 0$ μs). Based on the resulting $\Delta t$-spike rate functions (Fig. 3b, $n = 17$ neurons from 12 animals), we analyzed the change in spike rate during binaural stimulation compared to excitation only. We focused our analysis on the "best ICI" of each neuron, i.e., the ICI that elicited maximal response rate modulation as a function of cTD (see Methods and Supplementary Fig. 1b). Across all recorded neurons, the re-centering procedure revealed that increases in spiking occurred exclusively when inhibition was leading excitation by 400 or 600 μs relative to functional coincidence (median increase: 120.6 and 16.3%, $P = 0.00007$ and $P = 0.003$ respectively; Wilcoxon signed rank test; Fig. 3c). Remarkably, this specificity in the relative timing between excitation and inhibition was not only maintained throughout the click train, but spike rate changes were more pronounced for clicks later in the train (Fig. 3d, $P = 0.01$ at $\Delta t = 600$ μs and $P = 0.002$ at $\Delta t = 400$ μs, Friedman's test).

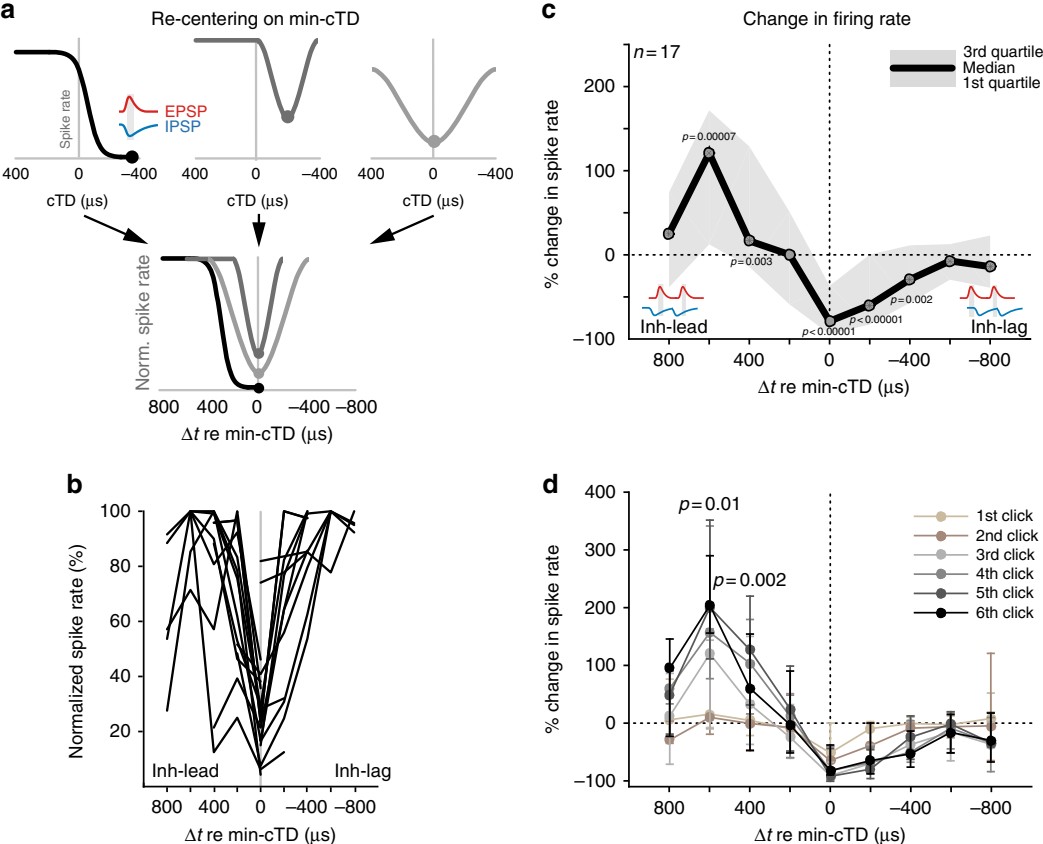

**Fig. 3** Precisely timed, preceding inhibition underlies spike facilitation in vivo. **a** Schematic of the re-positioning of cTD-spike rate functions: functions were re-centered on each neurons' min-cTD (illustrated by filled circles, only three schematic functions are shown here), resulting in an extended Δt-axis (−800 to 800 μs). This way, all cTD-response functions were normalized to the cTD of coincidence between excitation and inhibition (inset in top left panel). **b** Normalized Δt-spike rate functions at best ICI of each neuron (n = 17) after re-centering. **c** Median changes in spike rate relative to their respective response rate during Exc. only stimulation for the re-centered Δt-spike rate functions of all cells tested (n = 17). Next to the expected decrease of spike rates near Δt = 0, an increases in relative spike rate is apparent for Δt values of 600 μs (120.6%, P = 0.00007, paired Wilcoxon signed rank test) and 400 μs (16.3%, p 0.003, paired Wilcoxon signed rank test). Shaded area denotes interquartile ranges. **d** A similar dependency of response facilitation on Δt is apparent on the level of changes in firing rates to individual clicks (color-coded). Notably, facilitation increased with clicks later in the train (P = 0.01 and P = 0.002 at Δt = 600 μs and Δt = 400 μs, respectively, Friedman's test). Data points are mean ± s.e.m

In addition to changes in the number of spikes, we also observed that spike timing significantly shifted as a function of Δt (Supplementary Fig. 2a, b; n = 15 neurons from 11 animals). In particular, compared to the timing during excitation only, spikes occurred significantly earlier when inhibition led excitation by 600 μs (median: 96.6 μs, P = 0.002; Wilcoxon signed rank test;) to 800 μs (median: 85.7 μs, P = 0.0006; Wilcoxon signed rank test;), coinciding with the delay of highest spike facilitation (Fig. 3c). We also detected a reduction of the standard deviation ("jitter") of spike timing during binaural stimulation (Supplementary Fig. 2c, d). Notably, these various effects of timed inhibition on spiking were present across the range of ICIs tested (Supplementary Fig. 3).

**Inhibition facilitates spiking and alters its timing in vitro.** To gain a mechanistic understanding of the facilitation and spike-timing modulation, we developed an acute in vitro brain slice preparation in adult gerbils (postnatal days 31–38), and examined temporal interactions between excitation and inhibition at LSO somata using conductance-clamp. To this end, we measured excitatory and inhibitory synaptic conductance waveforms in voltage-clamp to obtain conductance templates that reflect natural synaptic kinetics in adult gerbil LSO neurons

(Supplementary Fig. 4a–b). We then injected calculated currents corresponding to these conductance templates given the instantaneous membrane voltage relative to a set reversal potential for excitation (+5 mV) and inhibition (−85 mV). This method of introducing synaptic events to the soma is advantageous over simple somatic current injections in that it more accurately recapitulates the membrane voltage in response to synaptic events. To assess whether the fundamental phenomenon observed in vivo can be reproduced in our in vitro preparation, we started by applying six-event stimulus trains of excitatory synaptic conductance templates at an ISI of 5 ms with a fixed amplitude for each event ($G_e$) to mimic the six click trains performed in vivo (Fig. 4). Because AP probability was very sensitive around the excitatory conductance threshold, we delivered excitatory templates in 1 nS steps around this threshold (35 ± 2.7 nS). While an AP on the first event was consistently elicited, repetitive spiking throughout the train was rarely observed (Fig. 4a, b, gray traces) and in some cases failed to produce subsequent APs even at the highest $G_e$ tested (3–5 nS above threshold, n = 3). We next paired the excitatory conductance with a corresponding inhibitory conductance train with a fixed amplitude ($G_i$, 50 nS) at relative time differences (rTDs) spanning an entire event cycle (0.25 ms steps from 0 to −4.75 ms). Inhibition at certain rTDs faithfully reduced spiking (Fig. 4a, red traces), but facilitated spiking at

other rTDs and furthermore permitted spiking throughout the train (Fig. 4b, red traces). These descriptive results are already reminiscent of the timing dependence of excitation and inhibition observed in vivo as well as the tendency for activity at the inhibitory ear to preferentially affect spikes later in the click train (Fig. 3d).

Indeed, plotting AP probability as a function of rTD revealed a tight time window for facilitation only during the repolarization phase of inhibition. However, this PIF quickly saturated at higher $G_e$ values (Fig. 4c, yellow area with red outline) and this saturation effect was particularly strong for the first two events. To facilitate comparisons to the in vivo data, where spike rates

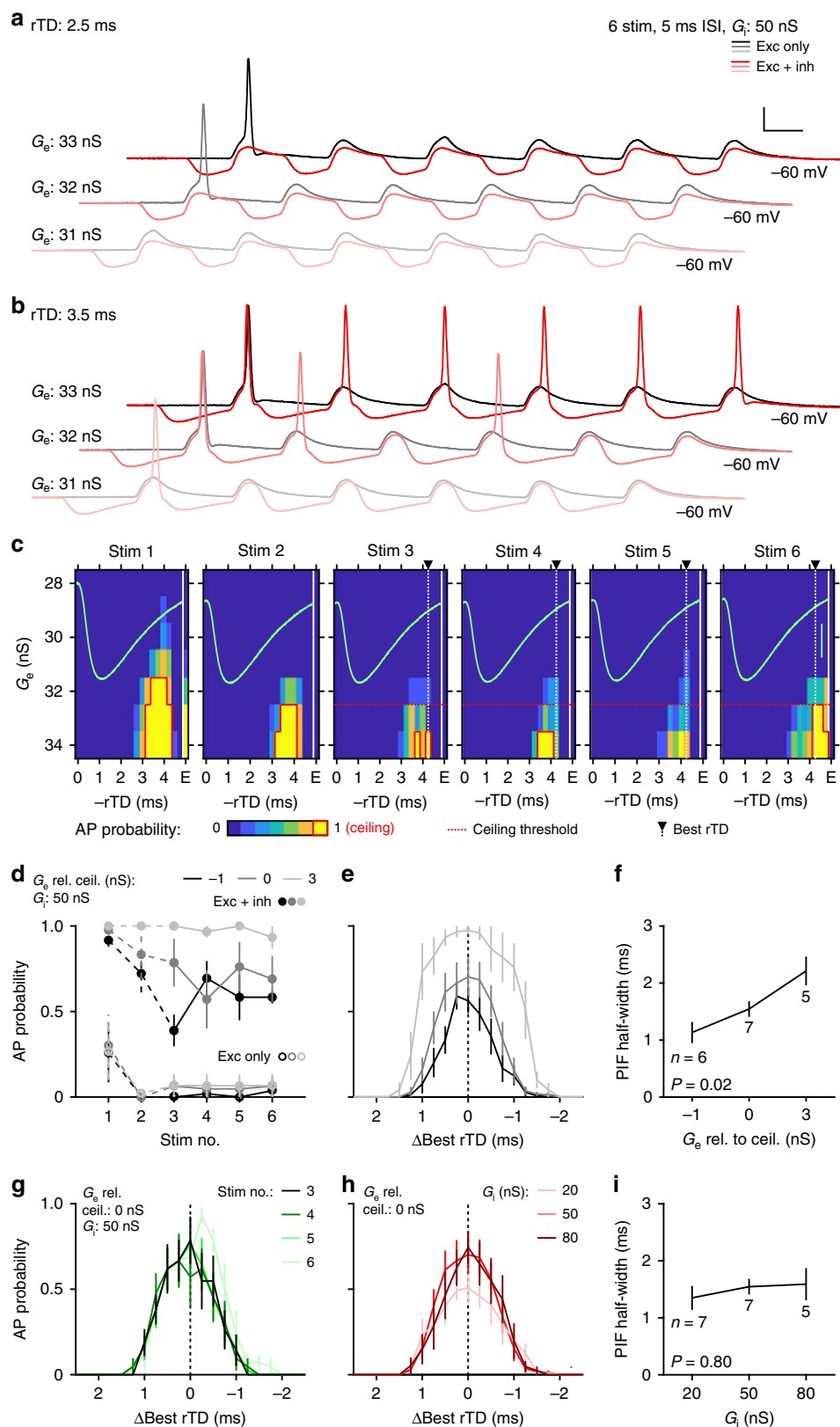

increased more moderately without saturation (Fig. 2d), we grouped the in vitro recordings according to their corresponding $G_e$ values that generated a ceiling effect ($G_e$ rel. to ceil., see Methods). Furthermore, because in vivo spike rate enhancement was only statistically significant on the last four clicks (Fig. 3d), we concentrated our in vitro analyses likewise on the last four events, to allow the best possible comparison to our in vivo findings. Plotting population-averaged AP probability (at each neuron's best rTDs, $-3.71 \pm 0.21$ ms, $n = 9$) for each event in the train illustrates the $G_e$ sensitivity (Fig. 4d). A $G_e$ value of just $+3$ nS above saturation caused 100% spiking on nearly every event, whereas the corresponding excitation-only condition produced nearly no APs beyond the first event. Plotting corresponding PIF functions (averaged across the last four train events) revealed the exquisite sensitivity of the timing window of PIF on $G_e$ (Fig. 4e). Specifically, at 1 nS below saturation the PIF function half-width was just $1.13 \pm 0.18$ ms ($n = 6$), whereas at $+3$ nS this value increased to $2.21 \pm 0.25$ ms ($P = 0.02$, Kruskal–Wallis test, $n = 5$, Fig. 4f). In contrast, at a given $G_e$ rel. to ceil. condition (0 nS), PIF functions for each of the last four events were extremely stable (Fig. 4g), and changing the strength of inhibition did not significantly influence PIF function half-widths ($P = 0.8$, Kruskal–Wallis test, Fig. 4h, i). Thus, PIF in vitro in the mature LSO constitutes a sizeable and robust phenomenon. Moreover, its timing window matches well the in vivo counterpart of spike rate enhancement, particularly for ongoing events (Figs. 3d and 4g, h).

We went on to determine whether other effects of inhibition on excitation that we had observed in vivo were evident in vitro as well. For example, the finding that spike rates were anti-correlated with changes in spike timing (Supplementary Fig. 2) predicts that timing conditions that boost firing in vitro should likewise advance spiking, whereas those that suppress firing should delay spiking. For this line of investigation, we considered only a single event and $G_e$ values that generated an AP without inhibition (Fig. 5a). This allowed us to calculate the difference in AP timing imposed by inhibition. We also quantified here the magnitude of inhibitory boost (in nS) based on the corresponding $G_e$ values that generated 50% AP probability (Supplementary Fig. 5a, b). As already apparent from single traces (Fig. 5a, right), we found timing conditions that advanced AP timing and others that delayed it ($P = 6.53 \times 10^{-6}$, one-way ANOVA, Supplementary Fig. 5c). Pooling data from all recordings (irrespective of rTD) against the magnitude of facilitation showed a small, yet statistically significant inverse correlation between facilitation of spiking and modulation of AP timing ($r = -0.64$, $P = 0.0034$, Supplementary Fig. 5d). We additionally examined whether inhibition increased AP timing precision as it did in vivo (Supplementary Fig. 3) and found a general suppression of AP jitter (Supplementary Fig. 5e, f). Thus, PIF in vitro successfully recapitulated important in vivo effects of inhibition on spike timing.

What biophysical mechanism could be underlying PIF? One possibility is that the hyperpolarization itself effectively relieves

$Na_v$ channels from inactivation, as evidenced by a decrease in the voltage AP threshold ($V_t$) enforced by inhibition (Fig. 5a, right, and Fig. 5b). To test this possibility, we mimicked an IPSP waveform in voltage-clamp and at varying times during the IPSP brought the neurons abruptly to a suprathreshold potential to evoke an inward AP current (Fig. 5c). This revealed a subtle but reliable increase in peak current (Fig. 5d, bottom), which faithfully followed the time-course of the IPSP (Fig. 5d, top). Across the data set, we observed a significant relative increase in inhibition induced AP current (Fig. 5e, bottom, $P = 0.002$, Wilcoxon signed rank test), albeit with considerable variability in the magnitude of boosts (median = 9.9%, interquartile range = 14.7%, $n = 10$ cells). These findings suggest that despite the hyperpolarization evoked by inhibition, a relief from $Na_v$ channel inactivation lowers the effective AP voltage threshold and could explain the ability of inhibition to promote repetitive spiking throughout our six-event trains.

Another possible biophysical contributor to PIF is the hyperpolarization-activated cyclic nucleotide-gated (HCN) channel, whose current ($I_h$) promotes rebound spiking activity in the neighboring superior paraolivary nucleus[45,46]. Although the PIF we report here is not a rebound spike per se (because it requires excitation), we evaluated the involvement of $I_h$ in PIF (Fig. 5e–i and Supplementary Fig. 5g–j). Indeed, blocking $I_h$ currents with the selective antagonist ZD 7288 (ZD) revealed a robust current with a fast-activating time constant of just 43 ms. Because the $I_h$ block also hyperpolarized the membrane potential and increased membrane resistance ($41.5 \pm 7.3$ M$\Omega$ compared to $17.4 \pm 2.0$ M$\Omega$, $n = 7$ cells, $P = 0.01$, paired $t$-Test, Supplementary Fig. 5h, i), we injected positive currents to bring neurons back to their previous resting membrane potential and found that after these manipulations synaptic postsynaptic potentials (EPSPs and IPSPs) on average exhibited similar kinetics to control conditions (EPSPs: $0.55 \pm 0.10$ ms vs. $0.51 \pm 0.09$ ms, respectively, $P = 0.6$; IPSPs: $3.31 \pm 1.21$ ms vs. $2.74 \pm 0.51$ ms, $P = 0.67$, $n = 4$, paired $t$-Test, Fig. 5e, f). We then compared single-event PIF protocols under baseline conditions and after wash-in of ZD (Fig. 5g–i, $n = 4$). Although ZD did influence AP shape, PIF persisted in the absence of $I_h$ (Fig. 5g). In fact, maximum inhibitory boost (Control: $2.00 \pm 0.81$ nS; ZD: $4.70 \pm 1.63$ nS, $P = 0.18$, paired $t$-Test) and PIF function half-widths (Control: $1.55 \pm 0.31$ ms; ZD: $2.28 \pm 0.64$ ms, $P = 0.32$, paired $t$-Test) tended to be larger in the presence of ZD (Fig. 5i), indicating that, although HCN channels are not required for the expression of PIF, they may work to sharpen PIF tuning. Finally, we also investigated the kinetic determinants of PIF and found a strong dependence of best rTD and inhibitory boost on cellular input resistance and membrane kinetics, respectively (Suppl. Fig. 6).

**PIF boosts ILD coding**. What might be the functional significance of a facilitation of weak excitatory inputs by preceding

**Fig. 4** Inhibition modulates spike generation in vitro. **a**, **b** Voltage traces from an example conductance-clamp recording in response to six-event trains with an inter-stimulus interval (ISI) of 5 ms. Excitation alone generated at most an onset spike (gray traces) at all excitatory conductances ($G_e$s) tested. The same protocol performed with an identical inhibitory train [peak conductance ($G_i$) = 50 nS] at a relative timing difference (rTD) of −2.5 ms suppressed spiking (**a**, red traces), but at an rTD of −3.5 ms promoted spiking and recruited spiking throughout the train (**b**, red traces). Scale bar: 20 mV, 2 ms. **c** Event-wise spike probability heat maps as a function of $G_e$ vs. rTD for the experiment shown in **a**, **b** with corresponding IPSP trace overlaid (green, scale bar: 5 mV). Red border outlines $G_e$ conditions that resulted in 100% spike probability on more than one rTD (ceiling). Dotted red line indicates the ceiling threshold for this recording. Spike probability from excitation alone (**e**) is shown at the right of each plot. Dotted white line and arrowhead indicates the calculated "Best rTD" for this recording. **d** Event-wise plot of spike probability for pooled data (at Best rTD) for $G_e$ values relative to the value generating a ceiling effect ($G_e$ rel. ceil.) and with $G_i$ = 50 nS. **e** Population average spike probability as a function of rTDs relative to the Best rTD for events three to six. Trace intensity reflects the $G_e$ rel. ceil. values in **d**. **f** Post-inhibitory facilitation (PIF) function half-widths for each $G_e$ rel. ceil. value tested. **g** Event-wise, population averaged spike probability at ceiling threshold $G_e$ and $G_i$ = 50 nS. **h** Same as in **e**, but for ceiling threshold $G_e$ and different $G_i$ values. **i** Same as in **f**, except for all $G_i$ values tested. All averaged data represent mean ± s.e.m

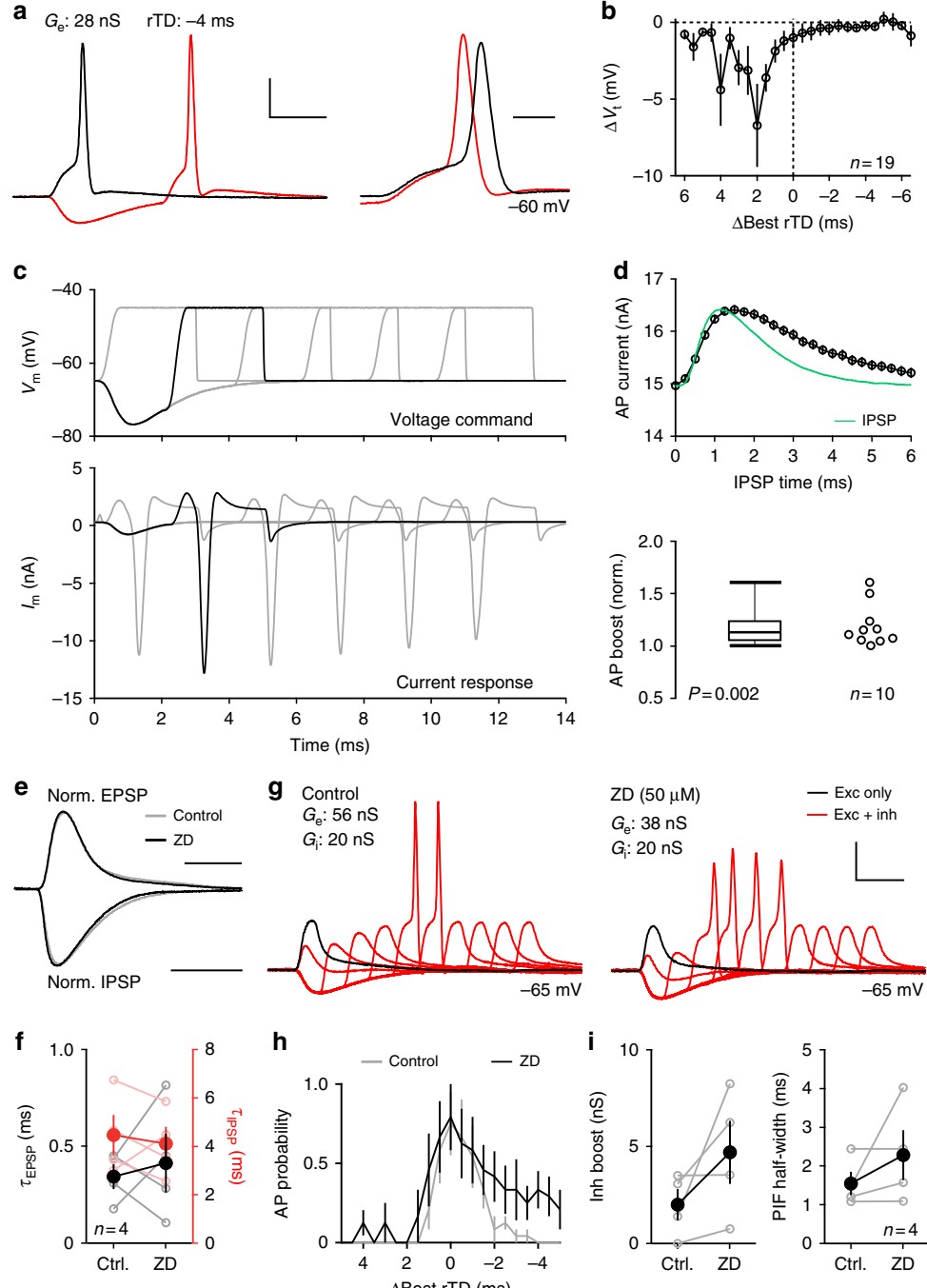

**Fig. 5** Changes in membrane excitability and HCN channel activity contribute to PIF in vitro. **a** Voltage traces from an representative single-event conductance-clamp experiment at a $G_e$ value that generated a spike from excitation alone (black). The same protocol performed with an inhibition time at –4 ms rTD (red) also produced a spike but occurred earlier and with a lower voltage threshold ($V_t$, right). Scale bars: 20 mV, 2 ms (left) and 0.5 ms (right). **b** Quantification of the relative change of spike voltage threshold as a function of ΔBest rTD. **c** Example traces from a voltage-clamp recording where the voltage command (top) started as an IPSP waveform, but then stepped to a suprathreshold value (3 ms duration) after a particular time (IPSP time). The resulting current response (bottom) shows a transient inward current that reflects sodium currents associated with an AP. Traces at an IPSP time of 2.5 ms are highlighted in black. **d** Top: mean AP current amplitude as a function of IPSP time for the recording in **c**. Inverted IPSP time course (green) is overlaid to compare with the time course of AP current modulation. Bottom: box plot shows the median and first/third quartiles (whiskers denote the range) of the distribution of AP boost for all recordings (open markers). **e**, **f** Example normalized traces (**e**) and quantification (**f**) of EPSP and IPSP decay kinetics before and after HCN channel blocker (ZD) wash-in. Scale bars (**e**): 2 ms (top) and 5 ms (bottom). **g** Example traces from a single-event conductance-clamp PIF protocol before (left) and after (right) ZD wash-in. Scale bar: 20 mV, 2 ms. **h** Average PIF functions for before (gray) and after (dotted black) ZD wash-in. **i** Population maximum inhibitory boost (left) and PIF function half-width (right) compared between control and after ZD wash-in. Light and bold lines and markers in **f**, **i** indicate individual experiments and population averages, respectively

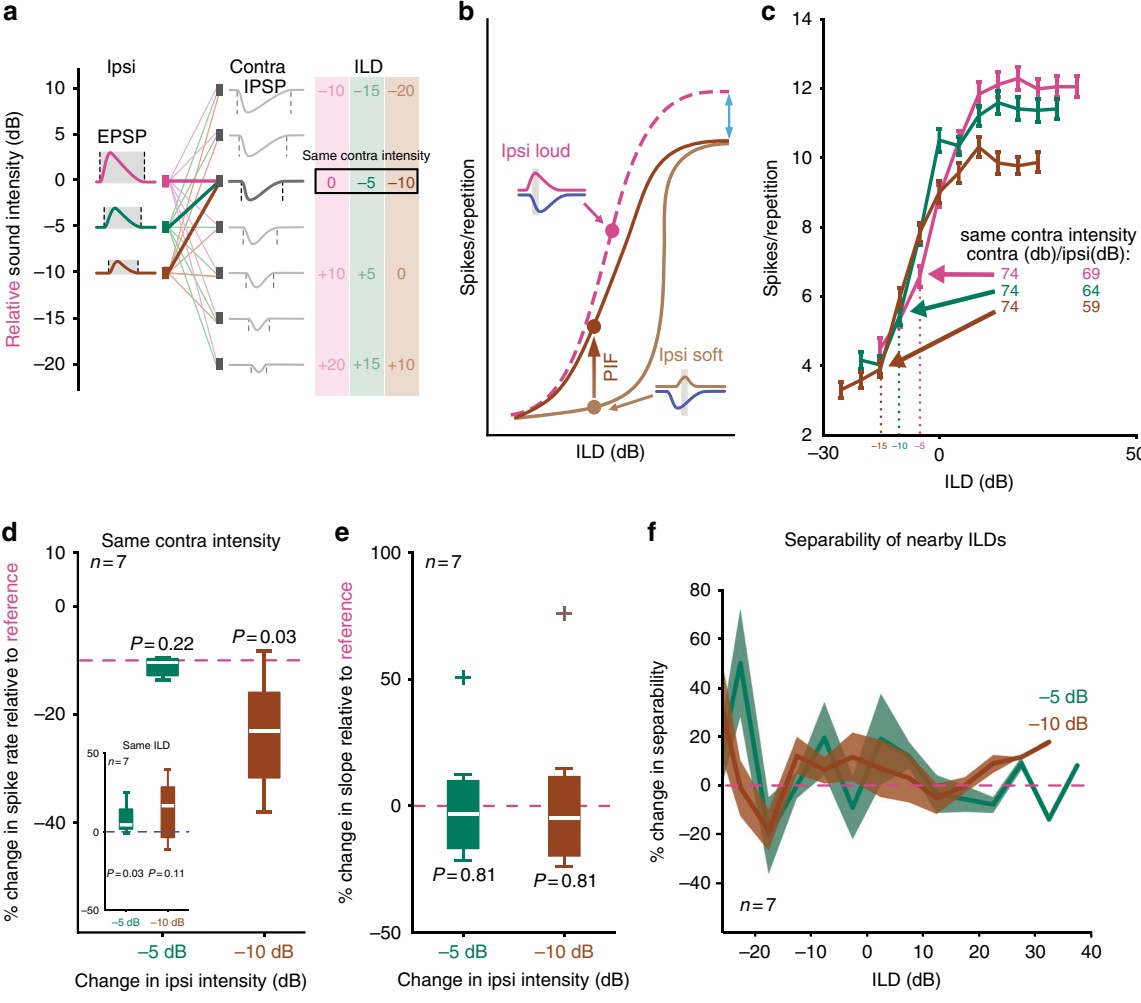

**Fig. 6** A functional role of post-inhibitory facilitation in the LSO. **a** Schematic of the stimulus design. Left: three ipsi sound intensities, normalized to the loudest (magenta ≙ 0 dB) were combined with a wide range of contra intensities. **b** Schematic of hypothetical ILD-response functions during soft (light brown) or loud (magenta) ipsilateral stimulation. Soft ipsi stimulation not only lowers spike rates at ipsi-favoring ILDs (blue arrow), but also increases the relative potency of the inhibition due to both amplitude and timing effects, resulting in a diminished dynamic range (steeper slope) under soft ipsilateral starting conditions. However, higher contra intensities are likely to cause the IPSP to precede the EPSP and thus generate PIF, which in turn can recover the dynamic range of the ILD-response function by an increase in spike probability along the slope (dark brown). **c** ILD-response functions of an example neuron (CF: ~16.2 kHz) in response to the three different ipsi conditions (color-coded) as introduced in **a**. Data are presented as mean ± s.e.m. **d** Changes in spike rates along the slopes of the ILD-response functions were only slightly affected by decreased ipsi intensities (−5 dB: −0.37% median change, interquartile range −2.75–0.24%; $P = 0.22$, Wilcoxon signed rank test; −10 dB: 13.0% median change, interquartile range −21.5 to −5.9%; $P = 0.03$, Wilcoxon signed rank test, $n = 7$ neurons). Inset: spike rates at identical ILDs were increased during reduced ipsi intensities (−5 dB: 4.4% median change, interquartile range: 1.9–14.3%, $P = 0.03$; −10 dB: 16.6% median change, interquartile range: −3.5–28.4%, $P = 0.11$, $n = 7$ cells). White horizontal bars show medians, interquartile range is given by box size; whiskers extend to most extreme data points. **e** Slope steepness remained unaltered between the three ipsi conditions (−5 dB: −3.1% median change, interquartile range: −16.6–9.8%, $P = 0.81$; −10 dB: −4.7% median change, interquartile range −19.6–11.4%, $P = 0.81$, $n = 7$ cells). Same conventions as in **d**. **f** Neuronal information about the separability of nearby ILDs was similar across ipsi conditions ($P > 0.05$ for all ILDs, $t$-test; solid lines and shaded areas show mean and s.e.m., respectively)

inhibition? We hypothesized that PIF is involved in maintaining the neuronal coding range (the slope of the ILD-response functions) over a range of stimulus intensities by boosting responsiveness to weaker signals. Without PIF, one would assume that for contralateral stimulus locations, the large increased inhibitory inputs would effectively prevent any spiking arising from the weaker excitatory inputs, resulting in a steepening of the slopes of the ILD-spike rate functions and effectively diminishing the dynamic range of the neuron. PIF may counteract this inhibitory dominance exactly at ILDs at which weak excitation is paired with stronger—and thus faster and putatively preceding—inhibition (Fig. 6a, b; compare to Fig. 1d). We tested this hypothesis by performing in vivo LSO recordings and pairing fixed

inhibitory intensities with progressively weaker excitatory intensities (Fig. 6a). The resulting ILD-response functions exhibited a high robustness to lowering the excitatory intensity (Fig. 6c, d). Specifically, when the excitatory intensity was lowered by 5 dB, response rates to a particular inhibitory intensity remained similar (−0.37% median change, interquartile range −2.75–0.24%; $P = 0.22$, Wilcoxon signed rank test, $n = 7$ neurons, Fig. 6d). If lowered by 10 dB, rates decreased only slightly (−13.0% median change, interquartile range −21.5 to −5.9%; $P = 0.03$, Wilcoxon signed rank test, $n = 7$ neurons, Fig. 6d). Most crucial to the neuronal coding capacities of the neurons, the slope of ILD-response functions was not significantly altered when the intensity on the excitatory ear was

decreased (−5 dB: −3.1% median change, interquartile range: −16.6–9.8%, $P = 0.81$, Wilcoxon signed rank test; −10 dB: −4.7% median change, interquartile range −19.6–11.4%, $P = 0.81$, Wilcoxon signed rank test, $n = 7$ neurons; Fig. 6e). This high slope robustness resulted in similar separability of nearby ILDs irrespective of the excitatory intensity (Fig. 6f, standard separability measure, see Methods and figure legend for details). Thus, PIF in the LSO likely serves to preserve the dynamic range of spatial sensitivity for ILD coding across sound intensities, suggesting that the microsecond relative timing of inhibition and excitation represents a fundamental mechanism of binaural integration for sound localization.

## Discussion

We determined a fundamental divergence from the typically assumed subtraction mechanism in the interaction between glutamatergic excitation and glycinergic inhibition in binaural auditory brainstem neurons. Synaptic inhibition decreased or increased the spike rate of LSO neurons depending on the particular sub-millisecond time difference relative to excitation. The increase was mediated by PIF to otherwise sub-threshold excitatory inputs, thereby maintaining high ILD sensitivity over a wide range of sound intensities. Contrary to current models of inhibitory function for processing microsecond differences in input timing[23,47,48], we provide direct evidence that in the intact brain, this high temporal acuity of inhibition is not limited to the first spike. Rather, it is precisely maintained and effective throughout click-trains. Whole-cell in vitro recordings reproduced several features of PIF and provided mechanistic insight into the underlying biophysical properties.

Previous studies showed that the effectiveness of inhibition to suppress spiking in response to short stimuli in LSO cells is restricted to specific phases of the IPSP[7,8,33,36,38]. Our observation that this temporal specificity of inhibitory effectiveness was maintained throughout the click train even at 500 Hz demonstrates its ability influence excitation on a cycle-by-cycle basis for relatively high click frequencies. These data thus challenge previous hypotheses on the role of MNTB-mediated inhibition for spatial processing, which suggested that inhibition was insufficiently fast to interact in a timing-dependent manner and rather functioned as a constant suppressor of excitation[23,26,48]. Our patch clamp data suggests that the response facilitation via inhibition is generated by hyperpolarizing the membrane potential prior to arrival of the excitatory inputs, which increased the likelihood of this input to trigger a spike. PIF is supported by a prior report of OFF-responsiveness in the LSO[49], and we have established that it is at least partially mediated by a reduction in the voltage threshold for spike initiation[50–52].

By varying stimulus settings, we found the temporal width of PIF in vitro to be highly sensitive to the event number (single event vs. train) and the applied excitatory conductance. This highlights a delicate balance in the interplay of inhibitory and excitatory inputs and the recruitment of associated cellular mechanisms that is required for PIF. Nonetheless, when matching both the stimulus paradigm (six-event train) and analysis, we observed a close match of the facilitation time window between the in vivo and in vitro data.

The entire mechanism of the facilitation we report here is likely to be complex, as we have shown that additional ion channels and conductances might modulate PIF. In particular, $I_h$ currents, which are involved in the classical post-inhibitory rebound spiking after long-lasting hyper-polarization[45,46], were too slow to underlie the facilitation itself, but may modulate its timing window. Contrary to post-inhibitory rebound spiking, PIF has, to our knowledge, so far only been observed in vitro in the MSO of juvenile mice. In these experiments, the deactivation of low-threshold potassium channels[43,53–55] resulted in response facilitation by leading inhibition[42]. Previous MSO data also explained how the interplay between synaptic inhibition and low-threshold potassium channels sharpens the temporal sensitivity of the binaural integration mechanism, particularly during ongoing activity[26,56].

The effects of inhibition on synaptic integration we report here have important implications for its role in binaural processing in the auditory brainstem. We found evidence that the inhibitory input can increase the separability of ILDs for weak stimuli. Human ILD separability has been shown to be robust against absolute sound level[57–59], and consequently PIF represents a potential underlying mechanism.

Under natural free-field conditions, ILDs and ITDs are co-modulated. For example, a transition of a sound source position from the ipsilateral to the contralateral hemisphere results in changes of both the ILD and ITD to favoring the contralateral ear. Consequently, the relative timing of the inhibitory input to the LSO will speed up because (a) the increase in sound intensity shortens the neuronal latency, and (b) the relative traveling time of the sound (i.e., ITD) is shorter to the inhibitory ear. For high frequency sounds, where ILDs are large[18], intensity (and thus latency)-dependent effects will dominate the input timing at the LSO relative to the ITD-dependent changes (Fig. 1d). However, the influence of ITDs will be more prominent for lower frequencies, where ILDs are rather small[18]. Conceptually, this consequently links the LSO and MSO, because they share synaptic inputs and the LSO is the likely evolutionary ancestor of the MSO[19]. It is thus tempting to speculate that the spike-timing-sensitive integration mechanism of the LSO might have served as a pre-adaptation for ITD processing in the MSO, and thus reflects the cellular blueprint for how the precise differences in the arrival of inputs tune peak excitation timing and consequently ITD sensitivity. Supporting this conclusion, in vivo MSO studies have repeatedly reported non-linear binaural integration, in which max-rates diverged from the sum of the monaural response rates[17,23,27,60]. Given that inhibition precedes excitation in the MSO by a few hundred µs[26], the response facilitation mechanism demonstrated here may be the cause for this non-linearity, because in vivo facilitation in the LSO occurred exactly when inhibition functionally led by 400 to 600 µs.

Next to response facilitation, we also provide evidence that both the absolute spike timing of LSO neurons and its jitter are significantly modulated by the relative arrival time of inhibition and excitation in vivo and in vitro, paralleling previous MSO in vitro results[24,26]. In particular, we showed that synaptic inhibition tunes ITD sensitivity in MSO cells by shifting the effective timing of the excitatory inputs[24]. Thus, our in vivo findings further strengthen the functionality of these mechanisms in the intact sound localization circuit.

Together, we have shown that fast synaptic inhibition originating from the MNTB provides important functions during binaural processing other than a reduction of spiking. Its evidently exquisite temporal precision persists after the onset of stimulation and profoundly affects crucial aspects of neuronal spatial sensitivity.

## Methods

**Ethical approval for animal experiments**. All experiments were approved in accordance with the stipulations of the German animal welfare law (Tierschutzgesetz) (55.2-1-54-2532-53-2015). Animals were housed in groups of 4 to 6 individuals with 12 h light/dark cycles.

**In vitro LSO recordings**. LSO slice preparation and electrophysiology: In vitro experiments were conducted on male and female Mongolian gerbils (*Meriones unguiculatus*) aged 31–38 days (19 animals). Gerbils were anesthetized with iso-flurane [2-chloro-2-(difluoromethoxy)-1,1,1-trifluoro-ethane] (IsoFlo, Zoetis Inc.) and decapitated. Brains were removed and placed in ice-cold dissecting solution containing (in mM): 93 $N$-methyl-D-glucamine, 93 HCl, 30 NaHCO$_3$, 25 glucose, 20 HEPES [4-(2-hydroxyethyl)-1-piperazineethanesulfonic acid], 10 MgCl$_2$, 5 L-ascorbic acid, 3 *myo*-inositol, 3 Na-pyruvate, 2.5 KCl, 1.2 NaH$_2$PO$_4$, 0.5 CaCl$_2$ (pH 7.4 when oxygenated with 95% O$_2$ and 5% CO$_2$). Two hundred micrometer-thick transverse brainstem slices were cut using a VT1200S vibratome (Leica). They were incubated at 35 °C for 30 min in dissecting solution and for another 30 min in an oxygenated perfusion saline containing (in mM): 125 NaCl, 25 NaHCO$_3$, 25 glucose, 3 *myo*-inositol, 2.5 KCl, 2 Na-pyruvate, 2 CaCl$_2$, 1.25 NaH$_2$PO$_4$, 1 MgCl$_2$, and 0.4 L-ascorbic acid (pH 7.4). Recordings were performed in perfused oxygenated saline (1 ml min$^{-1}$). Temperature was measured near the slice and maintained at 35 ± 1 °C by an SF-28 in-line heater (Warner Instruments) and a PH-1 bath chamber heater (Biomedical Instruments). Tissue was visualized under an Axios-kop upright microscope (Zeiss) equipped with infrared Dodt gradient contrast optics. Whole-cell recordings from principal neurons visually confirmed to be within the LSO were obtained with borosilicate glass electrodes using an EPC10/2 patch-clamp amplifier (HEKA Elektronik).

Synaptic stimulation: For synaptic stimulation recordings (Supplementary Fig. 4), the internal solution contained the following (in mM): 122 Cs-MeSO$_4$, 10 HEPES, 10 Na$_2$-phosphocreatine, 5 QX-314 [N-(2,6-dimethylphenylcarbamoylmethyl) triethylammonium chloride], 4 MgCl$_2$, 4 Na$_2$-ATP, 3 Na-L-ascorbate, 0.2 Cs-EGTA (cesium ethylene glycol tetraacetic acid), 0.4 Na$_2$-GTP, and 0.03 Alexa Fluor 633, adjusted to pH 7.25 and 297 mOsm. Voltage-clamp recordings at –70 mV were made with electrodes of 2–3 MΩ tip resistances, and series resistance (4–7 MΩ) was compensated to a residual of 1.5–2 MΩ on the amplifier. Experiments were terminated if the uncompensated series resistance changed by >10%. Synaptic inputs were stimulated with borosilicate glass electrodes (of 3–4 MΩ resistance) filled with saline and placed in the vicinity (50–150 μm) of the recorded neuron. Lateral (excitatory) and medial (inhibitory) inputs were activated every 2 s with brief (0.2 ms) 10–50 V bipolar pulses generated by a Model 2100 isolated pulse generator (A-M Systems).

Conductance-clamp: For conductance-clamp experiments, the internal solution contained the following (in mM): 145 K-gluconate, 15 HEPES, 5 Na$_2$-phosphocreatine, 3 Mg$^{2+}$-ATP, 0.3 Na$_2$-GTP, and 0.05 Alexa Fluor 592, adjusted to pH 7.25 and 320 mOsm. Current-clamp recordings were made with electrodes of 2–3 MΩ tip resistances, and series resistance (4–6 MΩ) was 100% balanced on the bridge of the amplifier. The liquid junction potential was estimated to be 15 mV and subtracted offline. All reported membrane voltages reflect this subtraction.

Selected synaptic conductance waveforms as measured in Supplementary Fig. 4a–c were delivered to an SM-1 conductance injection amplifier (Cambridge Conductance), which calculates instantaneous current commands [I($t$)] by equation (1).

$$I(t) = G(t)[V(t) - E_{rev}] \qquad (1)$$

This calculation was performed independently for excitatory [reversal potential ($E_{rev}$) = 5 mV] and inhibitory ($E_{rev}$ = –85 mV) synaptic conductance waveform templates [G($t$)], while simultaneously measuring the membrane potential [V($t$)].

Ramps (1 nS increments) of excitatory conductance ($G_e$) templates were applied to determine the spike (AP) conductance threshold. Then the same conductance ramps were performed in the presence of an inhibitory conductance ($G_i$) template of 20, 50 or 80 nS, varying the relative timing difference (rTD) of inhibitory to excitatory event onset. Templates were delivered as single events in 0.5 ms rTD steps between –1 to 10 ms (Fig. 5, and Suppl. Figure 4, $n$ = 19 recordings) or as six events summated at 5 ms inter-stimulus intervals (ISIs) in 0.25 ms rTD steps between 0 and 4.75 ms (Fig. 5a–c, $n$ = 9 neurons). These protocols were repeated six times for each condition tested.

Pharmacology: To evaluate a role of $I_h$ currents in PIF, the selective HCN channel antagonist ZD 7288 (Sigma) was applied (50 μM, $n$ = 7 cells). ZD-sensitive currents were measured using the $K$-based internal solution (as in Figs. 4 and 5) in voltage-clamp using a 500 ms step protocol from –50 to –80 mV. Because HCN channels are partially open at rest, blocking them hyperpolarized cells, requiring a 185 ± 99 pA current injection to bring the neurons back to their normal resting potential (–71.5 ± 1.2 mV for this subset of recordings). Under this compensation, membrane resistance (22.3 ± 4.2 MΩ) was similar to control conditions (19.3 ± 2.5 MΩ, $P$ = 0.24, paired $t$-Test). In a subset of recordings ($n$ = 4 cells), conductance-clamp PIF protocols (as in Fig. 4) were performed before and after wash-in of ZD.

AP current measurement: Using the same potassium-based internal solution and electrode specifications of the conductance-clamp experiments, neurons were held at their resting membrane potential (–67 ± 6 mV, $n$ = 10 recordings) in voltage-clamp. Using IPSP waveform templates approximated from conductance-clamp experiments in the same recording, a command template of a portion of the IPSP waveform that abruptly depolarized to a suprathreshold voltage (–45 to –15 mV) after 0 (no inhibition) to 10 ms of IPSP waveform (IPSP time) in 0.25 ms increments. This causes neurons to attempt an AP, which under our recording conditions was reflected by a sharp inward current. This protocol was repeated six times for each recording ($n$ = 10 recordings, a subset of the 19 recordings in Fig. 5).

IPSP waveforms in response to both inhibitory conductance values were evaluated, and data for these experiments were pooled across recordings.

In vitro data acquisition and analysis: Data were acquired on the EPC10/2 at 100 kHz. Voltage-clamp data were additionally filtered 8 kHz with a shallow three-pole Bessel filter. Analyses were performed offline in Igor Pro (Wavemetrics) or Matlab (The Mathworks) software. Example synaptic stimulation traces represent the average of 50–100 trials. Example conductance-clamp traces are raw traces. Example AP current traces are an average of six repetitions. Averaged data points indicate mean ± s.e.m, and linear correlation values were obtained from Pearson's correlation calculations).

For single event experiments, analysis of AP probability was performed at the largest $G_e$ ramp step that failed to return an AP on any trial for excitation alone. Because recordings exhibited diverse rTDs of maximal facilitation, the average AP probability function for each recording was fitted with a Gaussian, from which the peak was determined to be the Best rTD. For all subsequent analyses, rTD values were subtracted from this value. To quantify the magnitude of inhibition-induced facilitation, an AP conductance threshold was determined as the interpolated $G_e$ value that returned 50% AP probability. Values for each rTD were then subtracted by the corresponding value for excitation alone. To compare AP properties, analysis was performed at the $G_e$ ramp step, which produced an AP on at least one trial for excitation alone. AP timing and voltage threshold were determined at the peak time of the voltage double-derivative. For each rTD these values were subtracted from those obtained for excitation alone. AP jitter was determined as the s.d. of spike timing for all six at each rTD and was normalized to values obtained in excitation alone conditions. Data were only accepted for timing conditions in which minimally three spikes were elicited.

For six-event PIF experiments, analysis of AP probability was performed at $G_e$ ramp steps relative to the condition that caused 100% AP probability ("$G_e$ rel. to ceil.") for at least two rTDs on the last four events. This allowed an assessment of the influence of excitatory drive on PIF function half-width and restricted data analysis to the same clicks that exhibited PIF in vivo.

For AP current experiments, currents were calculated from averaged current traces (six trials) as peak-to-peak values. The IPSP time that returned the largest current was then normalized to IPSP time = 0 (no inhibition). Analysis was performed only on recordings where inhibitory boost was also measured.

**In vivo LSO recordings**. Surgical procedures: In vivo experiments were conducted on Mongolian gerbils (Meriones unguiculatus) 3–7 months of age and of either sex (15 animals). Gerbils were anesthetized by an intraperitoneal injection of ketamine (20%, Medistar GmbH) and xylazine (2%, Bayer AG) diluted in 0.9% NaCl solution (50 μl g$^{-1}$ body weight). Anesthesia was maintained with a continuous sub-cutaneous application of the same solution (2.4 μl per 100 g body weight per minute) using a syringe pump (Univentor Ltd). Anesthesia was routinely mon-itored by checking the hind leg withdrawal reflex.

The animal was placed on a thermostatically controlled heating pad (Fine Science Tools GmbH) to maintain the body temperature at 38 °C that was monitored using a rectal probe. The scalp was cut to reveal the dorsal part of the skull. Anterior to bregma, a metal rod was glued onto the skull using UV-sensitive dental-restorative material (Charisma, Heraeus Kulzer GmbH). The neck muscles at the recording side were partially removed to reveal the skull posterior to lambda. The animal was then transferred to a sound-attenuated chamber onto another thermostatically controlled heating pad on a custom-made stereotactic setup[61] and the head was fixated by the metal rod. An electrocardiogram was installed to monitor the heart rate. A small craniotomy was performed between bregma and lambda for the reference electrode. To enable access to the LSO, a craniotomy and durotomy was made behind the sinus transversus lateral to midline. In order to impede micro-bleedings and to prevent dehydration, the surface of the brain was regularly rinsed and covered with a physiological NaCl solution (0.9%). The head of the animal was stereotactically aligned relatively to lambda. The tragus on both ears was incised to ease access to the ear canal and custom-made electrostatic headphones were placed into the ear canals.

**Acoustic stimuli**. Frequency responses between 15 to 90 kHz were calibrated for each animal and each speaker. Acoustic stimuli were generated digitally, converted to an analog signal (RX6, Tucker Davis Technologies Inc.) at 200 kHz sampling rate, attenuated (PA5; Tucker Davis Technologies Inc.) and conveyed to the headphones. White noise bursts (duration 200 ms; rise/fall times of 5 ms) were presented monaurally to the ipsilateral ear to find responsive neurons. LSO neu-rons were identified by an EI response (stimulation of the ipsilateral ear evoking spiking and graded suppression of spikes by increasing amplitudes of the con-tralateral ear). ILD-response functions (data not shown) were recorded using a cassette of binaural correlated noise stimuli with varying ILDs for each side respectively (ipsilateral: 19–84 dB SPL; contralateral: 49–74 dB SPL). The CF of the neuron was determined audiovisually using tonal stimuli 20 dB above threshold. To quantify the temporal resolution of binaural processing, a train of six clicks with a single-click-duration of 50 μs was presented binaurally. These click-trains were presented at five different ICIs (5, 4 3, 2, 1 ms) and five different cTDs (generated by applied ITDs of –400, –200, 0, 200, 400 μs). All stimulus combinations were presented in a pseudo-randomized order. The relative intensity on each ear (i.e., the ILD) was individually adjusted for each neuron according to two measures of

responsiveness: (I) robust responsiveness during monaural ipsilateral stimulation, and (II) significant modulation of response rate by cTD during binaural stimulation (see below). The resulting mean ILD ± s.e.m. for the neurons included in the dataset was 0.7 ± 6.1 dB. To test the influence of changes in the ipsilateral intensity on ILD coding, response rates of 7 LSO neurons (three animals) to a white noise stimulus (50 ms duration) of various contralateral intensity were compared for three different ipsilateral intensities. Contralateral intensities were selected for each cell so that the resulting three ILD functions were positioned along the slope of the ILD-response function.

Extracellular single-cell recordings: APs of single LSO neurons were extracellularly recorded using glass electrodes filled with 5 units/µl horseradish peroxidase (Sigma-Aldrich Corp.) diluted in a 10% NaCl solution (resulting in tip resistances of ~8–12 MΩ). Using a motorized micromanipulator (Inchworm controller 8200, EXFO Burleigh Products Group) for remote control, the recording electrode was lowered into the brain tissue at an angle of 20°. Neuronal responses were measured by a pre-amplifier (Electro 705, World Precision Instruments), amplified (TOE 7607, Toellner Electronic), filtered (Hum Bug Noise Eliminator, Quest Scientific Instruments Inc) and delivered to the computer via a real-time processor (RP2, Tucker Davis Technologies Inc.). Here, neuronal responses were analyzed online with BrainWare (Jan Schnupp, Tucker Davis Technologies Inc.) allowing audiovisual control and refinements of the recordings. Single-neuron responses were isolated by visual inspection and online and offline spike sorting. A signal-to-noise ratio of the spike waveform of > 5 was required for recorded neurons to be included for further analysis.

Histology: To mark recording sites, horseradish peroxidase was deposited iontophoretically by applying a current of 1 µA for 8 min. After conclusion of an experiment, the animal was injected with a lethal dose of Narcoren (Pentobarbital 160 mg ml$^{-1}$) intraperitoneally (2 µl g$^{-1}$). The thorax was opened, and a cannula was inserted into the left ventricle while the right atrium was cut to cause blood efflux. The animal was perfused with Ringer-solution (containing NaCl (0.9%), heparin (100 µl ml$^{-1}$) and 5 mM PBS in H$_2$O) for 10 min followed by a perfusion with 4% paraformaldehyde (PFA in PBS pH 7.4) for another 10–25 min. For fixation, the brain was removed from the skull and incubated in 4% PFA for 1–2 days at 4 °C. After fixation, the brain was washed three times for 10 min in PBS (0.02 M) and was then embedded in 4% agarose to maintain stability during brain slicing. Coronal brainstem slices of 50–80 µm thickness were prepared. Labeling of the recording site was accomplished using a 3, 3′-diaminbenzidine (DAB) substrate kit for peroxidase (Vector Laboratories, Inc.). For counterstaining, an additional neutral red staining was accomplished. The brain slices were transferred onto glass objective slides and covered using DePeX (Serva Electrophoresis GmbH). Overview images of respective recording sites were acquired using an Olympus virtual slide fluorescence microscope (brightfield, 10× magnifications, Olympus BX61VS, Olympus Corp.). In total, 13 of 17 recording sites could be confirmed histologically. In the remaining cases, no distinct DAB counterstaining was found but recording sites could be reconstructed from the track of the recording electrodes.

In vivo data acquisition and analysis: Click-cTD analysis: We analyzed the recordings of 17 single neurons (in 12 animals) in the LSO with CFs between 15–36 kHz (see Supplementary Fig. 1). Data were analyzed using custom-made programs in Matlab (The MathWorks, Inc.). LSO neurons were included for further analysis if they had a mean spike rate of >1 spike per repetition (calculated over 20 repetitions) at the cTD/ICI combination which elicited the maximal spike rate and if they showed significant cTD sensitivity. The ICI which showed the largest modulation of the response rate by changing the cTD (relative to the standard deviation at the peak and the trough) was defined as the best ICI. Significance of cTD tuning was assumed if the mean response rate modulation (for all six clicks at best ICI) was at least two times larger than the standard deviation of the response rate during monaural stimulation at the same ICI. For population analysis, cTD functions at best ICI of each neuron were re-centered to its respective min-cTD, resulting in a prolonged relative cTD axis (Δt re min-cTD) from +800 to –800 µs (Fig. 3a, b). For the comparison of slope steepness (Fig. 5), ILD-response functions were fitted with broken stick (i.e., piecewise linear) regressions. To compare separability based on ILD-response functions, the standard separation D was calculated as previously described[62]:

$$D\_n = |mu\_n + 1 - mu\_n| \ / \ (sqrt(sigma\_n + 1 \times sigma\_n)),$$

where mu_n + 1 and mu_n are the mean values of the responses to two ILD values while sigma_n + 1 and sigma_n are their standard deviation. Depending on normality of the distribution, population average data are shown by the mean ± s.e. m., or the median and the 25 and 75% confidence intervals (interquartile range). Accordingly, parametric or nonparametric tests were used to determine statistical significances (see text and figure legends).

**Data availability**. The data that support the findings of this study are available from the corresponding author upon reasonable request.

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

## Acknowledgements

This work was supported by the Collaborative Research Center SFB870 of the German Research Foundation (DFG) to B.G. and M.P., the DFG Priority Program 1608 to M.P. and the Bernstein Center for Computational Neuroscience BCCN, Project C-T2) to B.G. We thank Dr. Boris Chagnaud for providing an in vitro setup and materials, Dr. Christian Leibold for discussions and Dr. Lars Kunz for assistance with statistical analyses.

## Author contributions

B.B. performed in vivo experiments and histology, analyzed the data and contributed to writing the paper. M.H.M. designed and performed in vitro experiments and pharmacology, analyzed in vitro data and contributed to writing the paper. N.M. designed and performed in vitro experiments. A.R.C. performed in vitro experiments. E.F. designed in vitro experiments and contributed to writing the paper. B.G. conceived the experiments and contributed to writing the paper. M.P. conceived and designed the experiments, analyzed the in vivo data and wrote the paper.

## Additional information

**Competing interests:** The authors declare no competing interests.

