## [Peer Review File · Nature Communications]

Editorial Note: This manuscript has been previously reviewed at another journal that is not operating a transparent peer review scheme. This document only contains reviewer comments and rebuttal letters for versions considered at Nature Communications. Reviewer #2 was added during the peer review at Nature Communications.

Reviewers' comments:

Reviewer #1 (Remarks to the Author):

The authors have extensively revised their manuscript and have addressed most of the suggestions and concerns raised by the reviewers. As a result the clarity of the study has significantly improved.

Despite the authors' explanations, I am still confused about the functional significance of interaural time differences in the range of 400 to 600 us, which of course far exceeds the possible delta ts that can occur in gerbils. The authors argue that these large dts account for intensity-dependent latency changes in both pathways. However, in figure 3, the ITD-spike rate functions were re-centered for each neuron to its min-ITD. This re-centering now reflects the coincidence of excitation and inhibition (dt=0 us). In my view, this re-centering should nullify any latency differences that may occur in both pathways and because the intensity is held constant in these experiments, new intensity difference-dependent latency changes do not occur. But even under these condition, a 'latency-independent' ITD (dt re min-ITD) of at least 400 us is necessary to produce the spike enhancing effects of inhibition. What do I miss here?

My other concern is the large discrepancy between the dTs that produce facilitation in vivo (400 us) and in vitro (>2 ms). This is an important issue, which in my opinion has not been sufficiently resolved. The authors provide two possible explanations. For one, they argue that this discrepancy may reflect the different approaches that were used to calculate PIF in vivo and in vitro and thus may reflect an iceberg effect (meaning that the actual window could be even larger in vivo?). But it is not sufficiently clearly explained why similar approaches could not have been used in both settings. Secondly, the authors argue that the in vitro situation may not reflect the in vivo situation because different numbers of inputs may have been activated in both situations, and we do not know how many fibers are activated by sound. However, because gerbil LSO neurons receive only a few (1-3) MNTB inputs, as the author's lab has shown previously, a few more experiments could address test whether shorter time windows in vitro are possible with different stimulation settings. The large discrepancy between the two temporal windows is a serious issue that should not be brushed away because it raise doubts whether the in vivo and the in vitro experiments investigate the same functional phenomenon and thus whether the mechanisms found in vitro are applicable to the in vivo situation.

Minor:

Page 8: "...the facilitation tended to increase from the first to the sixth stimulus (... , P=0.40, ..). Given the P value of 0.4 I think it is inappropriate to talk about 'trends'.

Page 18, methods: "neurons were held at their natural membrane potential in voltage clamp". The actual membrane potential of LSO neurons in vivo is completely unknown and the membrane potential in reported from slice experiments is determined by the recording conditions. Therefore, 'natural' is misleading and should be deleted.

Reviewer #2 (Remarks to the Author):

This is a review of the manuscript "Precisely Time inhibition facilitates action potential firing" by Beiderbeck et al.

The authors report on spiking responses in LSO neurons in response to combinations of of inhibitory

and excitatory inputs. Their main finding -- that strategically timed inhibition can promote firing in LSO cells (post-inhibitory facilitation, PIF) -- appears valid, as demonstrated by recordings in vivo and in vitro, and accompanied by appropriate statistical significance testing. The presentation is clear in most spots, and should be of great interest to researchers studying ILD processing by LSO cells. Moreover, and more generally, neuroscientists/theorists interested in how neuron's and circuits combine excitation and inhibition to encode external stimuli will likely appreciate this contribution.

I enjoyed the article -- I find it to be an interesting investigation that advances understanding of LSO physiology. As a general remark, I find that the authors have done well to respond to the careful critiques of previous reviewers. The focus on gerbil LSO, the pairing of in vitro and in vivo observations, the investigations into mechanism of PIF, and use of appropriate statistics, are all strengths of this manuscript.

I assess that, after revisions, the manuscript could be acceptable for publication in Nature Comms.

I detail my concerns below. Some comments follow up on critiques raised by reviewers of a previous version of this manuscript. I did not review earlier versions of this manuscript, but have been asked to comment on the author's responses to that prior round of review.

Line 58: "latency of input depends on the sound amplitude... changes in ILD consequently entail a change in arrival times". I agree that ITD & ILD co-vary in natural sounds, but these few sentences confuse me. Latency of input and sound amplitude at ear depend on sound source location, but saying "latency of input depends on sound amplitude" makes cause and effect unclear. Also in your reply to previous reviewer, you talk about other sources of delay (HRTF-related, conduction in neurons). These are important points in clarifying the rationale for your study/method. I see this language in the Results section, but perhaps it should be moved into the introduction to help clarify.

Line 97 - some typos - "enable to assess the temporal sensitivity of the binaural integration" . Possible change: "enable assessment of the temporal sensitivity of binaural integration"

Line 104 - typo - "spikel"

Line 125 and throughout - "effect of inhibition" in in vivo experiments.

It is clear from our understanding of LSO circuitry and physiology that contralateral inputs evoke inhibitory inputs to LSO neurons, but I would propose more "conservative" language. Specifically, "effect of addition of ipsilateral inputs" or something similar. In the in vitro setting you can control input type, but in the in vivo setting isn't it possible that there are other modulatory or "secondary" effects that may be unaccounted for, i.e. can you be 100% confident in the in vivo setting that playing sound to ipsilateral is equivalent to adding inhibition to an LSO cell?

Fig 1: somewhat related to the previous remark, and following up on a comment by Reviewer 3 in previous manuscript. In reply to that reviewer, you mentioned controls for "cross-talk" by recording monaural responses for both ears. This should be mentioned and/or shown in manuscript to confirm that ipsilateral-only inputs are not promoting spiking by some unanticipated mechanism.

Line 142: Suggested change: "These" to "Those". (When I read "These" I interpreted as referring to your own data in this manuscript, not data from MSO slice recordings by others).

Line 174-180: This reader did not immediately understand the differences & uses of synaptic waveforms, versus ramps. Please clarify here (not just in methods section).

Line 214: Why the claim that build-up of inhibition should suppress spikes". If the "operating point" of the neuron is the key, i.e. how much Na is available (via deinactivation) for spike generation, then this is not counter-intuitive is it?

Line 240: typo? remove "and" at start of line?

Line 289: 200Hz - 500Hz does not strike me as "high frequency", in context of ILD processing, or gerbil range of hearing.

Line 331-332: "Our findings thus provide insight into ... ITD coding as well". This sentence is unclear to me and may be unnecessarily provocative. It seems like the kind of "muddying of waters" between MSO and LSO to which previous reviewers objected, and that the authors have done well to clarify in this version of the manuscript. Your report is explicit about "ITD" (controlled time differences using click trains). so I don't understand the meaning of "as well" in line 332. One reading is that by "ITD coding as well" you are suggesting that your LSO results apply to MSO. But this seems 1) unnecessary and unjustified and 2) not entirely novel in the sense that PIF has been observed in MSO slice (as you remark) and also may shift "peak firing rates" in timing-specific manner, as in Myoga et al 2013.

Precisely timed inhibition facilitates action potential firing for spatial coding in the auditory brainstem

Barbara Beiderbeck, Michael H. Myoga, Nicolas Müller, Alexander R. Callan, Eckhard Friauf, Benedikt Grothe, and Michael Pecka

POINT BY POINT REPLIES

We thank both reviewers for their constructive comments and suggestions, which we have all addressed and incorporated in the new manuscript. These modifications substantially improved the comprehensibility of the rationale and the clarity of data presentation and interpretation.

Below are our point-by-point replies (in green) to the reviewers' comments.

Reviewer #1:

The authors have extensively revised their manuscript and have addressed most of the suggestions and concerns raised by the reviewers. As a result the clarity of the study has significantly improved.

Thank you.

Despite the authors' explanations, I am still confused about the functional significance of interaural time differences in the range of 400 to 600 μ s, which of course far exceeds the possible Δt s that can occur in gerbils. The authors argue that these large Δt s account for intensity-dependent latency changes in both pathways. However, in figure 3, the ITD-spike rate functions were re-centered for each neuron to its min-ITD. This re-centering now reflects the coincidence of excitation and inhibition ($\Delta t=0$ μ s). In my view, this re-centering should nullify any latency differences that may occur in both pathways and because the intensity is held constant in these experiments, new intensity difference-dependent latency changes do not occur. But even under these condition, a 'latency-independent' ITD (Δt re min-ITD) of at least 400 μ s is necessary to produce the spike enhancing effects of inhibition. What do I miss here?

The reviewer is correct in that the re-centering nullifies **respective** latency differences in our experiments, which can be different from cell to cell. The "normalization" of latency-differences (re-centering) to the functional coincidence between E and I allowed us to detect the high temporal specificity of facilitation across neurons.

Crucially, the "ITDs" or " Δt re min-ITD" we refer to were not meant to reflect naturally occurring ITDs that are produced by the inter-ear distance. Rather, the external ITDs were applied to systematically sample the effects of temporal differences between the two inputs on spiking within a range that can occur at LSO neurons under natural conditions. These differences are a combined product of changes in the inter-ear distance dependent ITD and, more prominently, the changes in latency for the inputs due to the changes in intensity at the ears that are associated with different sound source positions. For example, first-spike latencies in the auditory nerve typically change by approximately 1 millisecond for changes in the intensity by 10 dB (e.g Heil and Neubauer, 2001). Thus, a change in the location of a sound source from the ipsi- to the contralateral hemisphere will result in scenarios where the excitatory input is leading by hundreds of ms (for ipsi source locations) to a preceding of the inhibitory input by many hundreds of μ s (contra source location) (Yin et al. 1985, J Neurophysiol; Park

et al. 1996, J Neurosci.). Yet one can only determine the functionally specific effects of such timing differences when the intensity-associated amplitude changes that are normally also generated by changes in source location are omitted. We thus kept the ILD fixed and replicated the timing changes by applying delays in the presentation time on the two ears, which, in the pure physical sense, are ITDs. However, we realized how using the term “ITD” in figures 1-3 and throughout the manuscript, even though we consider it as physically correct, might be misleading, as these “ITDs” are meant to replicate the sum of the location-specific ITD and the latency difference at different sound source locations. We thus now refer to “cTDs” (composite timing delays) throughout the manuscript when referencing the applied delays on the two ears.

To further improve the comprehensibility of this rationale of timing changes and ILDs, we have now also refined the scheme of figure 1 and revised the Introduction and Results to explain the interplay of ILD/position-dependent changes in latency and ITD for input integration. To this end, we added a panel that more directly illustrates the effects that changing intensity on one ear has on input latency.

My other concern is the large discrepancy between the dTs that produce facilitation *in vivo* (400 μ s) and *in vitro* (>2 ms). This is an important issue, which in my opinion has not been sufficiently resolved. The authors provide two possible explanations. For one, they argue that this discrepancy may reflect the different approaches that were used to calculate PIF *in vivo* and *in vitro* and thus may reflect an iceberg effect (meaning that the actual window could be even larger *in vivo*?). But it is not sufficiently clearly explained why similar approaches could not have been used in both settings. Secondly, the authors argue that the *in vitro* situation may not reflect the *in vivo* situation because different numbers of inputs may have been activated in both situations, and we do not know how many fibers are activated by sound. However, because gerbil LSO neurons receive only a few (1-3) MNTB inputs, as the author’s lab has shown previously, a few more experiments could address test whether shorter time windows *in vitro* are possible with different stimulation settings. The large discrepancy between the two temporal windows is a serious issue that should not be brushed away because it raises doubts whether the *in vivo* and the *in vitro* experiments investigate the same functional phenomenon and thus whether the mechanisms found *in vitro* are applicable to the *in vivo* situation.

We agree with the reviewer that a thorough analysis of the relative timing for PIF between the *in vivo* and *in vitro* experiments is important, and we apologize if our discussion on this point was insufficient. Our primary motivation for the presented *in vitro* data was a proof of concept that preceding inhibition can facilitate spiking in mature LSO, and we were less concerned with replicating the exact time window observed *in vivo*. But the reviewer is correct that a more refined approach is advantageous.

To this end, we now provide new *in vitro* data and a refined analysis that is matched to our analysis procedure for the *in vivo* data. Doing so, we can now demonstrate that the timing *in vitro* is actually very similar to the timing observed *in vivo*. Specifically, by concentrating on six-event stimulation (mimicking the click train *in vivo*), we find a half width of the facilitation window for the ongoing events of ~ 1 ms, i.e. the dynamic range between maximal spike facilitation maximum and 0% change was only ~ 1 ms (new Fig. 4), only slightly larger than what we have observed *in vivo* ($\sim 600 \mu$ s, Fig. 3). This high temporal sensitivity was similar for each of the ongoing events (new Fig. 4g), providing further concordance with the single-click analysis in our *in vivo* observations (Fig. 3d).

To further examine the influence of stimulation settings (as requested by the reviewer), we now also provide new information how changes in the excitatory and inhibitory conductance (G_e and G_i , respectively) influence PIF, particularly the timing window. We show that the PIF half width is highly sensitive to the exc. G (new Fig. 4e-f), and can quickly saturate (new Fig. 4c-d), while the inhibitory G shows less of an effect (new Fig. 4h-i). We also performed a detailed analysis of the kinetic determinants of best rTD and inhibitory boost (new Suppl. Fig. 6).

We also would like to reiterate that not only spike facilitation in vitro resembled the in vivo results, but also the changes in jitter and spike latency were comparable, which further corroborates that the same phenomenon was studied in both preparations (Suppl. Figures 2 and 5).

Minor:

Page 8: “..the facilitation tended to increase from the first to the sixth stimulus (... , P=0.40, ..). Given the P value of 0.4 I think it is inappropriate to talk about ‘trends’.

We thoroughly revised this section of the manuscript, focusing almost entirely now on data from the six-event paradigm. Following from this, old figure panel 5c has been omitted.

Page 18, methods: “neurons were held at their natural membrane potential in voltage clamp”. The actual membrane potential of LSO neurons in vivo is completely unknown and the membrane potential in reported from slice experiments is determined by the recording conditions. Therefore, ‘natural’ is misleading and should be deleted.

We deleted “natural” and now use the term “resting membrane potential”.

Reviewer #2 (Remarks to the Author):

This is a review of the manuscript "Precisely Time inhibition facilitates action potential firing" by Beiderbeck et al.

The authors report on spiking responses in LSO neurons in response to combinations of of inhibitory and excitatory inputs. Their main finding -- that strategically timed inhibition can promote firing in LSO cells (post-inhibitory facilitation, PIF) -- appears valid, as demonstrated by recordings in vivo and in vitro, and accompanied by appropriate statistical significance testing. The presentation is clear in most spots, and should be of great interest to researchers studying ILD processing by LSO cells. Moreover, and more generally, neuroscientists/theorists interested in how neuron's and circuits combine excitation and inhibition to encode external stimuli will likely appreciate this contribution. I enjoyed the article -- I find it to be an interesting investigation that advances understanding of LSO physiology. As a general remark, I find that the authors have done well to respond to the careful critiques of previous reviewers. The focus on gerbil LSO, the pairing of in vitro and in vivo observations, the investigations into mechanism of PIF, and use of appropriate statistics, are all strengths of this manuscript.

I assess that, after revisions, the manuscript could be acceptable for publication in Nature Comms.

I detail my concerns below. Some comments follow up on critiques raised by reviewers of a previous version of this manuscript. I did not review earlier versions of this manuscript, but have been asked to comment on the author's responses to that prior round of review.

Thank you for your kind assessment and for making the extra effort to consider the prior round of reviews.

Line 58: "latency of input depends on the sound amplitude... changes in ILD consequently entail a change in arrival times". I agree that ITD & ILD co-vary in natural sounds, but these few sentences confuse me. Latency of input and sound amplitude at ear depend on sound source location, but saying "latency of input depends on sound amplitude" makes cause and effect unclear. Also in your reply to previous reviewer, you talk about other sources of delay (HRTF-related, conduction in neurons). These are important points in clarifying the rationale for your study/method. I see this language in the Results section, but perhaps it should be moved into the introduction to help clarify. We agree that clarification of this rationale is of utmost importance (see also response to reviewer #1). We have now further refined the scheme of figure 1 that explains the interplay of ILD/position-dependent changes in latency and ITD for input integration and have expanded the explanation both in the Introduction and Result section. To minimize confusion, we also changed our terminology: Instead of referring to "ITDs" when referencing the stimulation delays we applied to the ears, we now termed these "cTDs" (composite timing delay). Even though the term ITD would physically correct, it might be misleading, as these "ITDs" are meant to replicate the sum of the location-specific ITD and the latency difference at different sound source locations. Please see our reply to main concern #1 by the other reviewers and the tracked changes in the manuscript for details.

Line 97 - some typos - "enable to assess the temporal sensitivity of the binaural integration" . Possible change: "enable assessment of the temporal sensitivity of binaural integration"
Changed as suggested.

Line 104 - typo - "spikel"

Corrected.

Line 125 and throughout - "effect of inhibition" in in vivo experiments.

It is clear from our understanding of LSO circuitry and physiology that contralateral inputs evoke inhibitory inputs to LSO neurons, but I would propose more "conservative" language. Specifically, "effect of addition of ipsilateral inputs" or something similar. In the in vitro setting you can control input type, but in the in vivo setting isn't it possible that there are other modulatory or "secondary" effects that may be unaccounted for, i.e. can you be 100% confident in the in vivo setting that playing sound to ipsilateral is equivalent to adding inhibition to an LSO cell?

We were very careful during our recordings to check that stimulation of the contralateral ear indeed does not activate an additional excitatory input by "cross-talk" or other undocumented ways. We also would like to stress that the fact that enhancement and suppression of spiking in vivo was modulated by cTDs of only few hundred μ s is hard to reconcile with additional contralateral excitation or cross talk.

Nonetheless, we now provide additional control data to quantify the influence of the contralateral input in more detail. Please refer to our response to the following point.

Moreover, our *in vitro* data clearly show that the facilitatory effects can indeed be attributed to IPSCs, i.e. inhibition. Nevertheless, we agree that using a more conservative term when referring to the *in vivo* situation, especially before introduction of the *in vitro* results, might be advantageous. We thus corrected our use of the term "inhibition" in this context to now reading "contralateral ear /input" where applicable.

Fig 1: somewhat related to the previous remark, and following up on a comment by Reviewer 3 in previous manuscript. In reply to that reviewer, you mentioned controls for "cross-talk" by recording monaural responses for both ears. This should be mentioned and/or shown in manuscript to confirm that ipsilateral-only inputs are not promoting spiking by some unanticipated mechanism.

We provide new data providing direct evidence that stimulation of the contralateral ear did not recruit additional excitation (apart from PIF). To this end, we quantified the change in maximal spike rate in response to the click train when increasing the intensity on the contralateral ear by 5 dB (ipsi un-changed). The rationale is the following: If the enhancement of spiking that we observed would be caused by the recruitment of excitatory inputs (via cross-talk or undocumented contralateral excitatory projections), the increase in contra intensity should result in a further increase in maximal spike rates. However, in the subset of neurons for which data from this additional contra condition was available (n=9), the max spike rate actually decreased (both the overall max rate and the rate at the cTD at which PIF was observed in the control condition). These data thus not only provide clear evidence against the hypothesis of recruitment of contralateral excitation/cross-talk, but also demonstrate the delicate balance between relative amplitudes and timings for the ipsi- and contralateral inputs underlying PIF in the LSO.

In the new manuscript, we provide this new control data set as part of new figure 2

Line 142: Suggested change: "These" to "Those". (WHen I read "These" I interpreted as referring to your own data in this manuscript, not data from MSO slice recordings by others).

Changed as suggested

Line 174-180: This reader did not immediately understand the differences & uses of synaptic waveforms, versus ramps. Please clarify here (not just in methods section).

This section on the use of conductance clamp has been thoroughly revised and expanded, explaining that this method of introducing synaptic events to the soma is advantageous over simple somatic current injections in that it more accurately recapitulates the membrane voltage in response to synaptic events.

Line 214: Why the claim that build-up of inhibition should suppress spikes". If the "operating point" of the neuron is the key, i.e. how much Na is available (via deinactivation) for spike generation, then this is not counter-intuitive is it?

This section was completely revised because of the re-organization of the data and presentation of new data. The sentence that the reviewer referred to was omitted in the process.

Line 240: typo? remove "and" at start of line?

This section was completely revised because of the presentation of new data.

Line 289: 200Hz - 500Hz does not strike me as "high frequency", in context of ILD processing, or gerbil range of hearing.

We changed the sentence to now read: "Our observation that this temporal specificity of inhibitory effectiveness was maintained throughout the click train even at 500Hz demonstrates its ability influence excitation on a cycle-by-cycle basis for relatively high click frequencies."

Line 331-332: "Our findings thus provide insight into ... ITD coding as well". This sentence is unclear to me and may be unnecessarily provocative. It seems like the kind of "muddying of waters" between MSO and LSO to which previous reviewers objected, and that the authors have done well to clarify in this version of the manuscript. Your report is explicit about "ITD" (controlled time differences using click trains). so I don't understand the meaning of "as well" in line 332. One reading is that by "ITD coding as well" you are suggesting that your LSO results apply to MSO. But this seems 1) unnecessary and unjustified and 2) not entirely novel in the sense that PIF has been observed in MSO slice (as you remark) and also may shift "peak firing rates" in timing-specific manner, as in Myoga et al 2013.

We omitted this sentence to avoid unnecessary confusion.

REVIEWERS' COMMENTS:

Reviewer #1 (Remarks to the Author):

The authors did a great and thorough job addressing my comments. They revised the introduction and results and modified Figure 1. This makes it easier to follow their basic argumentations and helps to avoid possible misunderstandings. They also provide new in vitro data and new analysis to better link the in vitro results with the in vivo results. This greatly strengthens their main conclusions. My issues have all been resolved.

Reviewer #2 (Remarks to the Author):

This is a review of the revised manuscript "Precisely Time inhibition facilitates action potential firing" by Beiderbeck et al.

The authors have revised their previous manuscript and adequately addressed my concerns. The use of the new term "cTD" seems appropriate. The added data, figures, and explanation of conductance clamp methods improve the presentation. I think overall the manuscript is clear and an interesting contribution to the literatures of LSO physiology and excitation/inhibition interactions.

I have no further comments on this manuscript.

POINT BY POINT REPLIES

As is evident from the comments below, the referees were entirely satisfied by our revision and no further actions to modify the manuscript are required. We thank the referees for their continued effort.

Reviewer #1 (Remarks to the Author):

The authors did a great and thorough job addressing my comments. They revised the introduction and results and modified Figure 1. This makes it easier to follow their basic argumentations and helps to avoid possible misunderstandings. They also provide new in vitro data and new analysis to better link the in vitro results with the in vivo results. This greatly strengthens their main conclusions. My issues have all been resolved.

Reviewer #2 (Remarks to the Author):

This is a review of the revised manuscript "Precisely Time inhibition facilitates action potential firing" by Beiderbeck et al.

The authors have revised their previous manuscript and adequately addressed my concerns. The use of the new term "cTD" seems appropriate. The added data, figures, and explanation of conductance clamp methods improve the presentation. I think overall the manuscript is clear and an interesting contribution to the literatures of LSO physiology and excitation/inhibition interactions.

I have no further comments on this manuscript.